# Uber Movement Data: A Proxy for Average One-way Commuting Times by Car

**Yeran Sun [1,2], Yinming Ren [3] and Xuan Sun [4,5,*]**

[1]    Guangdong Provincial Key Laboratory of Urbanization and Geo-simulation, School of Geography and Planning, Sun Yat-sen University, Guangzhou 510275, China; sunyr8@mail.sysu.edu.cn

[2]    Department of Geography, College of Science, Swansea University, Swansea SA28PP, UK

[3]    School of Soil and Water Conservation, Beijing Forest University, Beijing 100083, China; Yinming.ren@bjfu.edu.cn

[4]    Zhou Enlai School of Government, Nankai University, Tianjin 300350, China

[5]    Computational Social Science Laboratory, Nankai University, Tianjin 300350, China

*    Correspondence: sunxuan@nankai.edu.cn

**Abstract:** Recently, Uber released datasets named Uber Movement to the public in support of urban planning and transportation planning. To prevent user privacy issues, Uber aggregates car GPS traces into small areas. After aggregating car GPS traces into small areas, Uber releases free data products that indicate the average travel times of Uber cars between two small areas. The average travel times of Uber cars in the morning peak time periods on weekdays could be used as a proxy for average one-way car-based commuting times. In this study, to demonstrate usefulness of Uber Movement data, we use Uber Movement data as a proxy for commuting time data by which commuters' average one-way commuting time across Greater Boston can be figured out. We propose a new approach to estimate the average car-based commuting times through combining commuting times from Uber Movement data and commuting flows from travel survey data. To further demonstrate the applicability of the commuting times estimated by Uber movement data, this study further measures the spatial accessibility of jobs by car by aggregating place-to-place commuting times to census tracts. The empirical results further uncover that (1) commuters' average one-way commuting time is around 20 min across Greater Boston; (2) more than 75% of car-based commuters are likely to have a one-way commuting time of less than 30 min; (3) less than 1% of car-based commuters are likely to have a one-way commuting time of more than 60 min; and (4) the areas suffering a lower level of spatial accessibility of jobs by car are likely to be evenly distributed across Greater Boston.

**Keywords:** Uber Movement; Travel time; Commuting time; Origin-destination Matrix; Aggregate data

---

## 1. Introduction

Travel time from one location to another has been widely used to measure transport accessibility [1–4]. Conventional travel survey data usually cannot collect accurate travel times since it highly relies on participants' memories. Widely used in transport research, Geographic Information System (GIS) provides an approach to accurately measure travel times [1–4]. However, travel times are usually estimated based on travel distance and average modal speed, but are not realistic [1–8]. For instance, some studies have accounted for locations of public transport services, locations of basic services, and road networks to estimate travel times by public transport [1,2,7]. Additionally, some other studies further take account of public transport service frequency to estimate travel times by public transport [3,4,6,8]. Although the majority of the relevant studies focused on the accessibility of public transport, some studies attempted to estimate travel times by private transport [5–7]. There are also some studies undertaking estimates of mode-based travel time to compare accessibility levels [5,7].

In recent years, the popularity of GPS-enabled devices has reshaped transport accessibility estimation. GPS-enabled devices not only help people to drive easily and safely but also produce massive GPS traces. Vehicle GPS data have great potential in complementing conventional travel survey data in transport research since they are spatio-temporally fine-grained and collected at a low cost. In particular, we can accurately infer the travel time between two locations from GPS traces. Moreover, vehicle GPS data enable individual-level transport research, whilst conventional travel survey data can only support aggregate-level transport research. GPS traces can provide detailed movement trajectories of vehicles in support of individual-level transport studies. Typically, among vehicle GPS data, floating car data (FCD) are likely to pave a new way for understanding individual-level travel behaviour and bring transport research to a big data era. In the last few years, FCD has been widely used in transport research, including the modelling of passenger demand [9–11], estimation of gas emissions [12–14], analysis of drivers' behaviour [15–17], and estimation of travel time [18–20]. There are several taxi FCD datasets open to the public, including Beijing datasets, New York City datasets, and so forth, and a number of studies have been undertaken around these open taxi FCD datasets [10,15]. At the same time, like other big data sources like social media, vehicle GPS data are attracting increasing privacy concerns. To protect user privacy, nowadays individual travel GPS traces are usually aggregated and anonymized before being publicly released for research and other purposes. Although some raw taxi GPS datasets are publicly available after being anonymized, raw GPS traces of other vehicles like private cars and bicycles are not publicly available due to privacy concerns. To address the conflict of research needs and privacy concerns, some institutions aggregate raw GPS traces of vehicles or persons to streets or small areas (e.g., neighbourhoods or census areas). Typically, commercial traffic data providers such as TomTom, Here, and Google also offer aggregated historical and real-time traffic information [21]. Some researchers have used real-time speed data from TomTom and Google to estimate dynamic accessibility or congestion level [21,22]. To exploit the power of citizen science, the crowdsensing approach has been recently used to collect travel data. Specifically, users are encouraged to self-report and share travel data via mobile devices [23–27]. Moreover, as a world-leading sports and fitness social media, Strava has released a data production named Strava Metro after aggregating individual GPS traces of Strava's users to the streets or census areas. Specifically, Strava Metro offers three types of aggregated dataset at the street, intersection and census area levels, respectively. Simply put, the counterparts of street- or intersection-level data are the conventional traffic count data, whilst the counterpart of the census area level data is the origin–destination (O–D) matrix.

Recently, as a world-leading car-sharing corporate, Uber also released datasets named 'Uber Movement' to the public in support of urban planning and transportation planning [28]. Likewise, to prevent user privacy issues, Uber aggregates Uber cars' GPS traces into small areas (e.g., census areas). Specifically, the average travel time of Uber cars between two small areas is computed by Uber. At this stage, the average travel times of Uber cars between two small areas by quarter, month, week, or hour of day are available. Uber cars' average travel times in the morning on weekdays could be used as a proxy for average one-way (home-to-work) car-based commuting times due to the same transport mode and time periods. Conventional travel survey data provide the amount of travel flows between census areas but rarely provide travel times between them. Specifically, conventional travel survey data are usually collected via telephone- or post-based questionnaires. Therefore, travel flow volumes, travel modes, travel times, and other travel-related information are all self-reported by the participants. Compared to other travel-related information, travel time cannot be well measured or accurately recorded based on participants' memories. Uber Movement data can provide more reliable and accurate car-based travel time information than conventional travel survey data.

Uber car driving speed might differ slightly from Uber ride-sharing cars and privately owned cars due to potential difference in the route selections and driving experience levels. In addition, travel times between two places might differ slightly from commuting trips and non-commuting trips. To make trips aggregated in the Uber Movement data be representative of commuting trips, we selected the data aggregated from Uber trips in weekday mornings to avoid the majority of non-commuting

trips. Although there might be still some non-commuting trips on weekday mornings, the travel time between the same origin and destination places might not differ much from commuting trips and non-commuting trips since the shortest or fastest routes are always the optimal routes selected by both commuters and non-commuters. Compared to travel mode and origin–destination, trip purpose is likely to have a relatively weak influence on travel time. To our knowledge, there are no published studies comparing Uber car driving speed and private car driving speed in reality. Since we can theoretically assume that Uber car driving speed is close to private car driving speed between the same origin and destination places during the same time periods, we attempt to use Uber Movement data as a proxy for one-way commuting times by car.

In this study, to demonstrate the usefulness of Uber Movement data, we use Uber Movement data as a proxy for commuting time data by which commuters' average one-way commuting time across Greater Boston can be figured out. We propose a new approach to estimate the average car-based commuting times through combining commuting times from Uber Movement data and commuting flows from travel survey data. To the best of our knowledge, this study is the first article to use Uber Movement data as a proxy for one-way commuting times. In the empirical study, we further compare the commuting times estimated from Uber Movement data and other data to somewhat validate the estimation. The remainder of this paper is organized as follows: Section 2 introduces the data and the methods used in this study, while later Section 3 carries out an empirical analysis, and finally, Section 4 presents the conclusion and offers suggestions for future work.

## 2. Materials and Methods

In this section, the Uber Movement data and travel survey data used are first introduced (see Section 2.1). Subsequently, we need to select the appropriate morning h and quarters to calculate annual mean travel times based on Uber Movement data (see Section 2.2). Finally, we estimate commuters' average one-way car-based commuting times (see Sections 2.3 and 2.4). Moreover, we will further present how to calculate spatial accessibility of jobs by car based on the place-to-place average one-way car-based commuting times.

### 2.1. Data

#### 2.1.1. Study Area

In this study, we chose Greater Boston as the study area. According to the American Community Survey (ACS), using averages, workers in Boston had a longer commute time (29.3 min) than the normal US workers (25 min) in 2016 [29]. As the average car ownership in Boston is 1 car per household, most people in Boston commute by driving alone [29]. In this case, commuting times by car can accurately represent the level of job accessibility in Boston.

#### 2.1.2. Uber Movement Data

Recently, Uber publicly released Uber Movement datasets for a number of cities worldwide [28]. To prevent privacy issues and keep spatio-temporally fine-grained information, Uber anonymized and aggregated Uber riders' GPS-tracked trails to census tracts or traffic analysis zones (or equivalent). There are a few US cities covered by Uber Movement datasets. Census tract and zip code are the main small area units that are supported by Uber Movement datasets. Compared to zip code, census tract is more widely used in US demographic surveys. Therefore, in this study, we chose census tract as the small area unit. The Uber Movement dataset for a city is a suite of aggregated datasets, including 'All-HourlyAggregate', 'All-MonthlyAggregate', 'OnlyWeekdays-HourlyAggregate', 'OnlyWeekdays-MonthlyAggregate', 'OnlyWeekends-HourlyAggregate', 'OnlyWeekends-MonthlyAggregate', and 'WeeklyAggregate'. Table 1 lists the attributes of the file 'OnlyWeekdays-HourlyAggregate', including 'Year', 'Quarter', 'Source ID' (origin census tract ID), 'Destination ID' (destination census tract ID), 'Hour Of Day' (a one-hour period) and 'Average Travel Time' (average travel time of Uber riders between origin

census tract and destination census tract during a one-hour period on the weekdays in a quarter of 2016). In Greater Boston, there are 1302 census tracts covered by Uber Movement data. Figure 1 maps census tract boundaries of Greater Boston.

**Table 1.** An example of records in the Uber movement datasets.

| Year | Quarter | Source ID | Destination ID | Hour Of Day | Average Travel Time (unit: second) |
|------|---------|-----------|----------------|-------------|-------------------------------------|
| 2016 | 1 | 63 | 237 | 6 | 567.77 |
| 2016 | 1 | 748 | 1189 | 10 | 938.54 |
| 2016 | 2 | 435 | 811 | 17 | 1273.05 |
| 2016 | 3 | 362 | 1016 | 20 | 1158.73 |
| … … | … … | … … | … … | … … | … … |

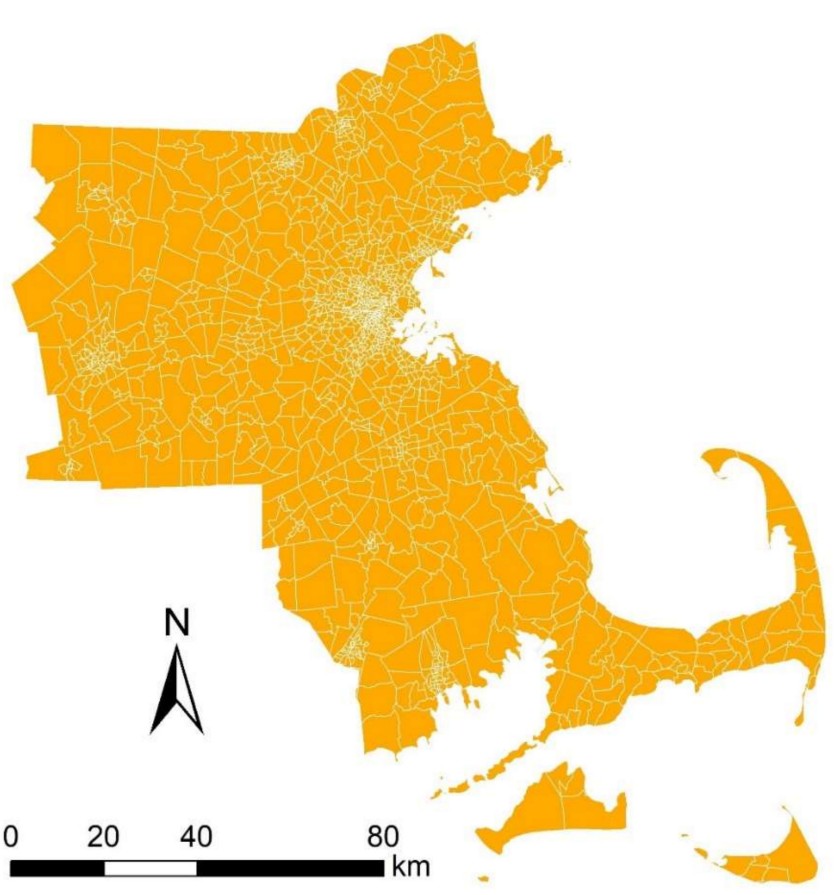

**Figure 1.** Census tract boundaries of Greater Boston.

### 2.1.3. Travel Survey Data: Commuting Flow Data

The O-D commuting flows data were downloaded from the U.S. Census Bureau [30]. As the Origin–Destination Employment Statistics (LODES) datasets offer the number of O–D commuting flows at the census block level, we aggregated O–D commuting flows into the census tract (CT) level. We used the data for 2015 as they are the most up-to-date data. Figure 2 boxplots census tract-to-census tract (CT-to-CT) number of workers (commuters) in Greater Boston. The majority of CT-to-CT number of workers (commuters) are likely to range from 2 to 5 persons across Greater Boston, whilst the largest number is over 1000.

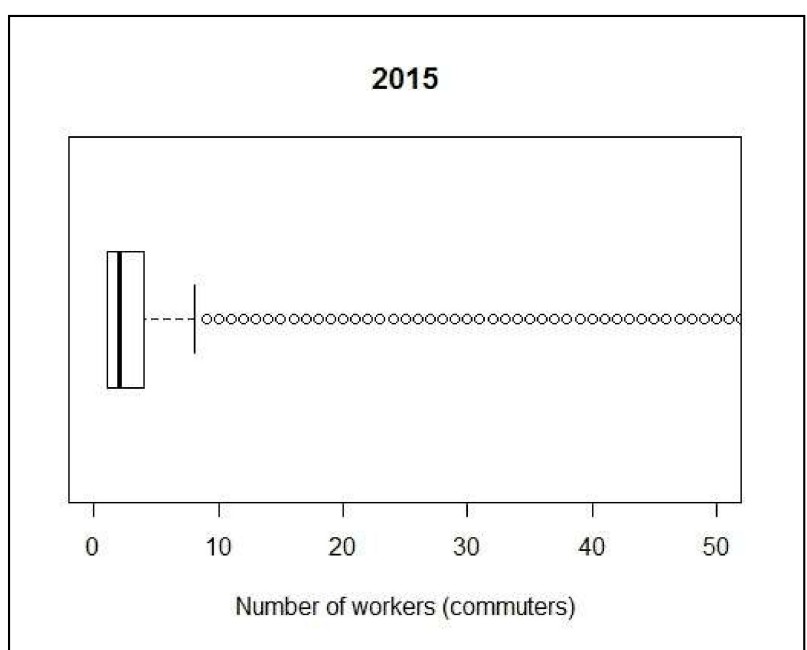

**Figure 2.** Boxplot of census tract-to-census tract (CT-to-CT) number of workers (commuters) across Greater Boston.

### 2.2. Temporal Analysis of Mean Travel Times

In this study, we first explored the distributions of small area-level mean travel times by hour of day and quarter. To do so, we found out the morning peak time periods and the appropriate quarters to calculate annual mean travel times.

### 2.3. Calculation of Annual Mean Travel Times

As the mean travel times of census tract pairs might vary over season, we needed to calculate the annual mean travel times of small area pairs by averaging the mean travel times of the four quarters.

### 2.4. Estimation of Commuters' Average Car-based Commuting Times

Since car-based commuters are unlikely to be distributed among census tract pairs, a population-weighted method is needed to calculate the average car-based commuting time. Accordingly, average car-based commuting times should be computed according to combination of average commuting times between small areas and number of car-based commuting flows between small areas. Theoretically, average travel time of a geographic region $R$ should be calculated as

$$Ave\_Time(R) = \frac{\sum_{i,\,j\,\in\,C} Ave\_Time\,(i,\,j) * Num\_Commuters\_By\_Car\,(i,\,j)}{\sum_{i,\,j\,\in\,C} Num\_Commuters\_By\_Car\,(i,\,j)} \tag{1}$$

where $Ave\_Time\,(i,j)$ is the average car-based commuting time between the two small areas ($i$ and $j$); and $Num\_Commuters\_By\_Car\,(i,j)$ is the number of commuters by car between the two small areas ($i$ and $j$). $i$ and $j$ are small areas within the geographic region $R$.

Equation (1) can be rewritten as

$$Ave\_Time(R) = \frac{\sum_{i,\,j\,\in\,C} Ave\_Time(i,\,j) * Num\_Commuters(i,\,j) * P\_By\_Car(i,j)}{\sum_{i,\,j\,\in\,C} Num\_Commuters\,(i,j) * P\_By\_Car(i,j)} \tag{2}$$

where $P\_By\_Car\,(i,j)$ is the proportion of commuters travelling to work by car between the two small areas ($i$ and $j$).

However, in some cases, *P_By_Car* (*i,j*) is unknown. If we assume *P_By_Car* (*i,j*) $\equiv$ *a* (*a* is a constant), then average travel time of a geographic region *R* can be calculated as

$$Ave\_Time(R) = \frac{\sum_{i,\,j\,\in\,C} Ave\_Time\,(i,\,j)*Num\_Commuters\,(i,\,j)}{\sum_{i,\,j\,\in\,C} Num\_Commuters\,(i,\,j)} \tag{3}$$

where *Ave_Time* (*i,j*) is the average car-based commuting time between the two small areas (*i* and *j*); and *Num_Commuters* (*i,j*) is the number of commuters between the two small areas (*i* and *j*). *i* and *j* are small areas within the geographic region *R*.

### 2.5. Estimation of Spatial Accessibility of Jobs by Car

Average commuting times of car users by origin small area is calculated to measure the spatial accessibility of jobs by car at the small area level. Theoretically, the spatial accessibility of jobs by car of a small area *i* should be calculated as

$$Job\_Accessibility(i) = \frac{\sum_{i,\,j} Ave\_Time\,(i,\,j)*Num\_Commuters\_By\_Car\,(i,\,j)}{\sum_{i,\,j} Num\_Commuters\_By\_Car\,(i,\,j)} \tag{4}$$

where *Ave_Time* (*i,j*) is the average car-based commuting time between the two small areas (*i* and *j*); and *Num_Commuters_By_Car* (*i,j*) is the number of commuters by car between the two small areas (*i* and *j*). *i* and *j* are small areas.

## 3. Results

In this study, we empirically estimated average one-way car-based commuting times of Greater Boston, and further identified the percentage of car-based workers who have a commuting time of less than 30 min. It is noted that we used Equation (3) instead of Equation (1) to calculate average car-based commuting times since the proportions of car-based commuters between census tracts are unknown.

### 3.1. Temporal Analysis of Mean Travel Times

We first explored the distributions of census tract-level mean travel times by hour of day (see Figure 3). As Figure 3 shows, in each quarter, the median mean travel times during Hour 7 (07:00–07:59) and Hour 8 (08:00–08:59) are longer than those during Hour 6 (06:00–06:59) and Hour 9 (09:00–09:59). Average speeds during Hour 7 and Hour 8 are lower than those during Hour 6 and Hour 9. This shows that Hour 7 (07:00–07:59) and Hour 8 (08:00–08:59) are likely to be the morning peak time period if a lower speed is associated with a higher volume of traffic. A statistical analysis also indicates that more than 40% of workers in the US leave home to go to work during Hour 7 (07:00–07:59) and Hour 8 (08:00–08:59) [31]. We then explored the distributions of mean CT-to-CT travel times by quarter (see Figure 4). As Figure 3 shows, in each morning hour, the median mean travel times of the four quarters are close to each other. In addition, all the mean CT-to-CT travel times of the quarters excluding Q1 are below 120 min (2 h). Moreover, we mapped the mean census tract-to-census tract (CT-to-CT) travel times during Hour 8 (08:00–08:59) or in Quarter 2 to show the temporal analysis of mean travel times. Specifically, Figure 5 shows the spatial distributions of mean census tract-to-census tract (CT-to-CT) travel times during Hour 8 (08:00–08:59) and Figure 6 shows the spatial distributions of mean census tract-to-census tract (CT-to-CT) travel times in Quarter 2. The two figures indicate that in the morning the predominant direction of longer commuting flows (e.g., more than 30 min) is towards the central area (Boston City).

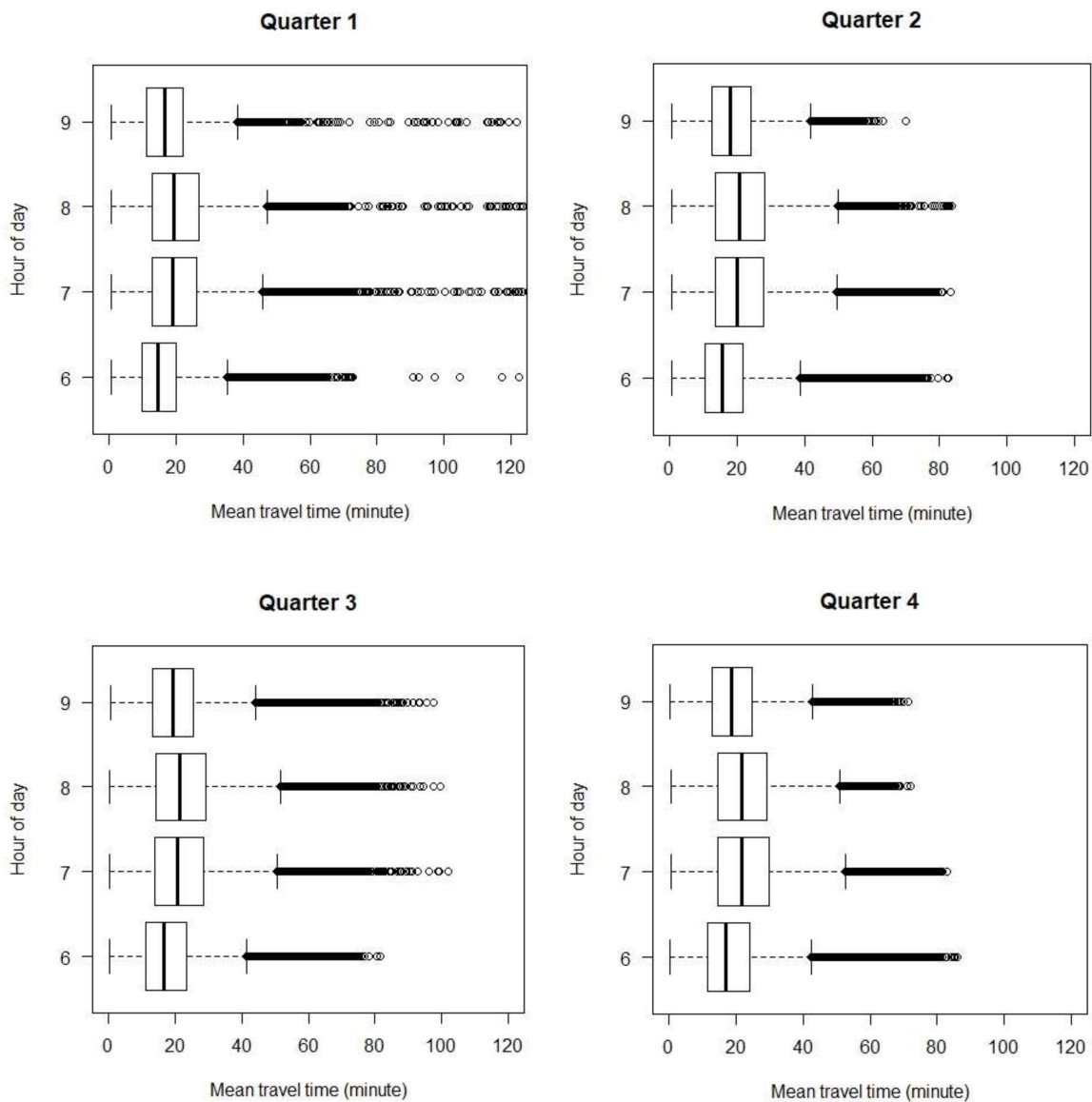

**Figure 3.** Distributions of mean census tract-to-census tract (CT-to-CT) travel times by hour of day.

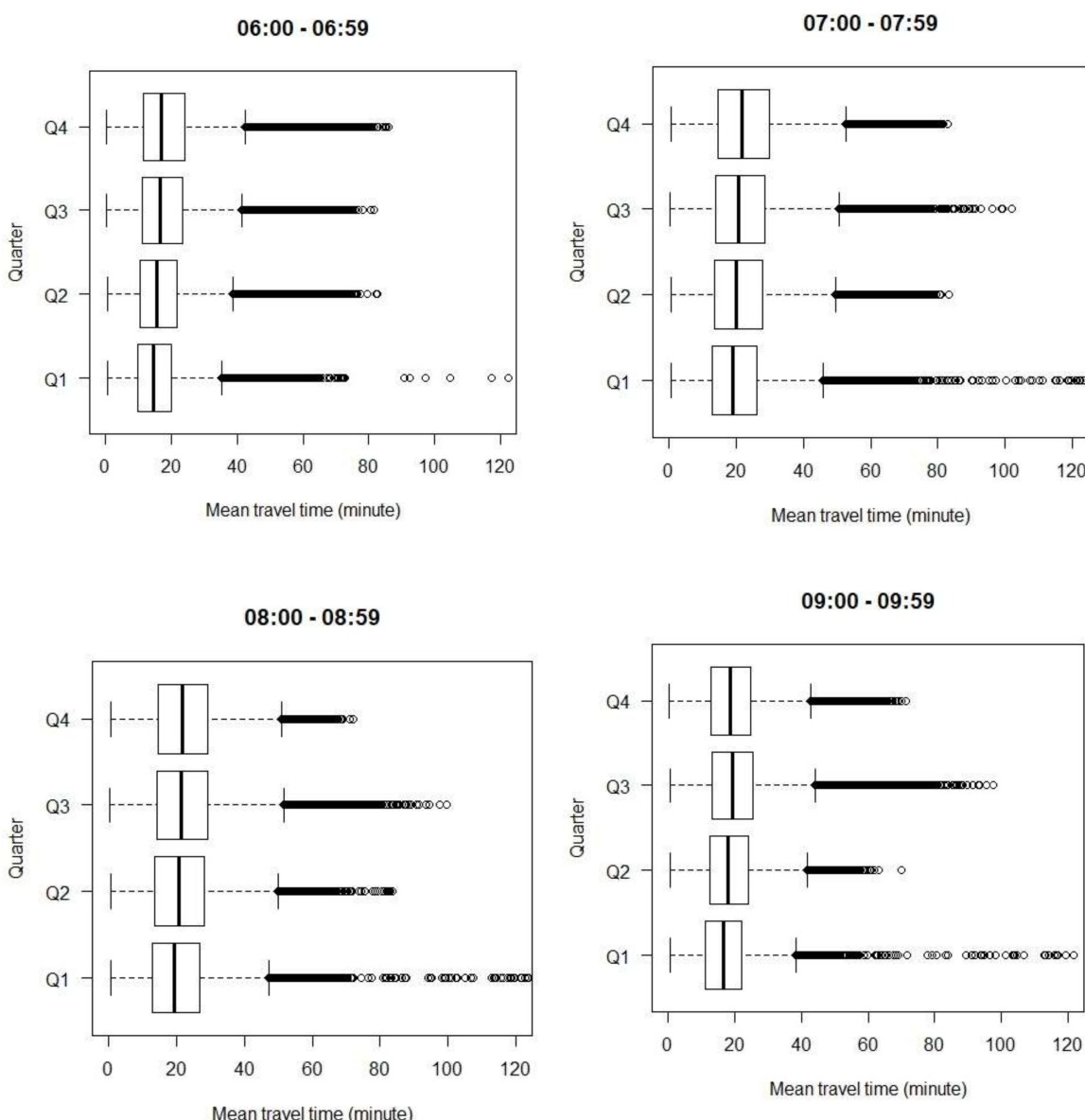

**Figure 4.** Distributions of mean census tract-to-census tract (CT-to-CT) travel times by quarter.

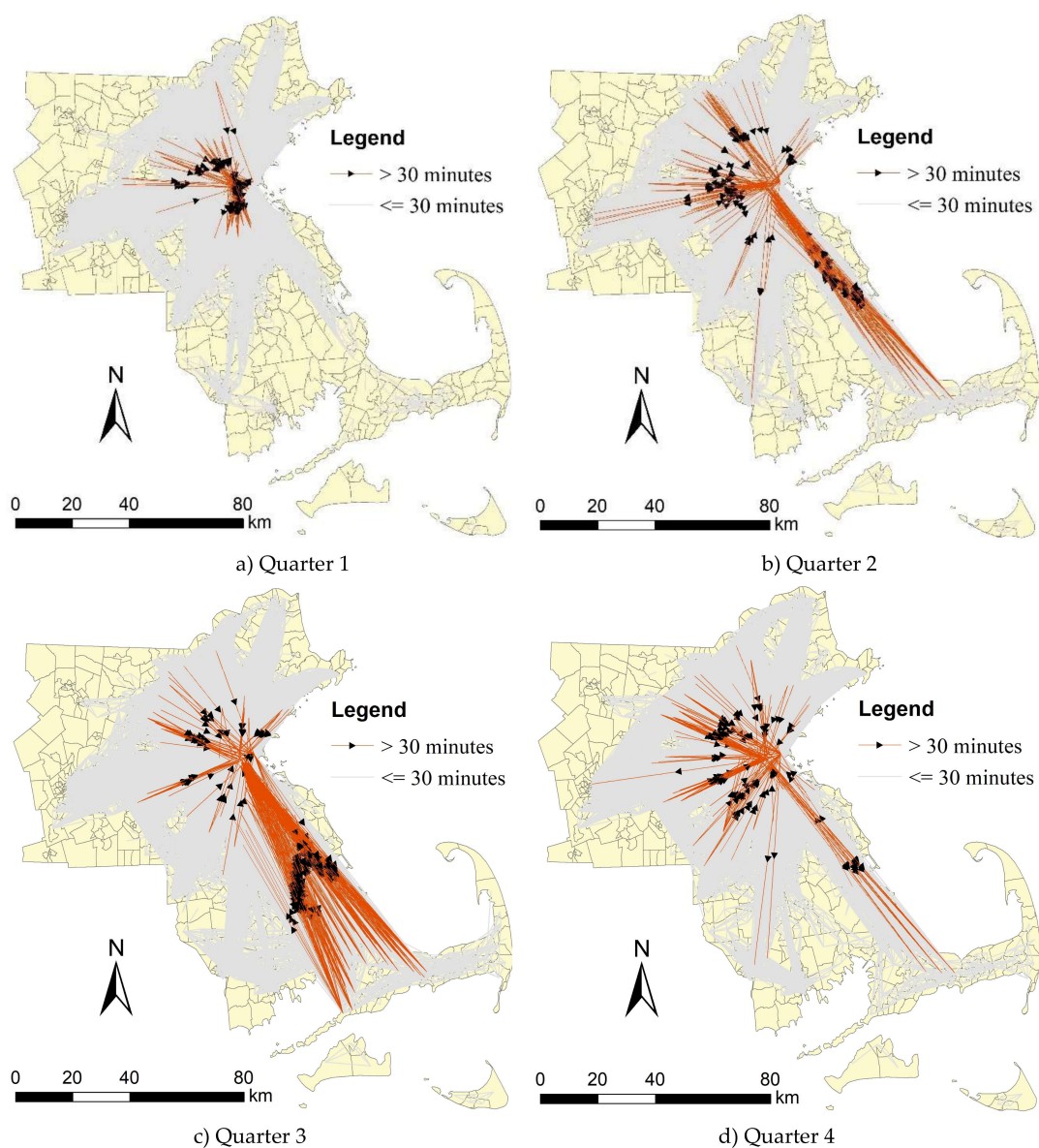

**Figure 5.** Spatial distributions of mean census tract-to-census tract (CT-to-CT) travel times during Hour 8 (08:00–08:59).

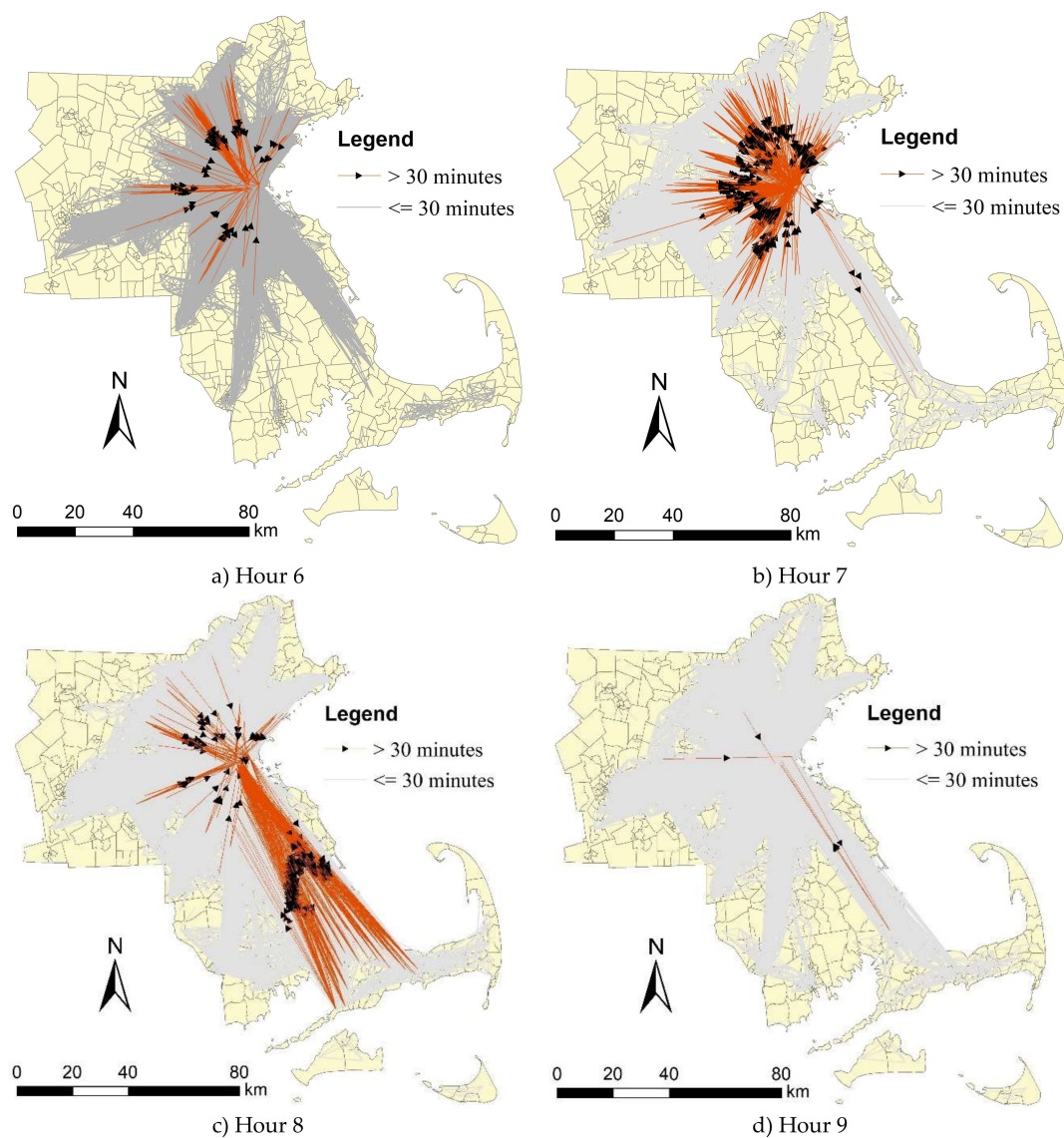

**Figure 6.** Spatial distributions of mean census tract-to-census tract (CT-to-CT) travel times in Quarter 2.

## 3.2. Calculation of Annual Mean Travel Times of Uber Cars

As Figure 3 shows, Hour 7 and Hour 8 are more likely to be the morning peak time periods due to the higher medians of mean travel times. In this study, we used the quarterly travel times of Uber cars during morning peak time periods (i.e., Hour 7 or Hour 8) to represent quarterly one-way car-based commuting times. Before calculating the annual mean travel times of Uber cars, we first needed to select the census tract pairs which have records in all the four quarters. As a result, 93,137 and 97,613 census tract pairs were selected for Hour 7 (07:00–07:59) and Hour 8 (08:00–08:59), respectively. Moreover, we used the coefficient of variation (CV) to measure relative seasonal variability for mean travel times of census tract pairs. The coefficient of variation (CV) is referred to as the ratio of the standard deviation to the mean. As Figure 4 shows that Q1 has a few outliers which are extremely high, we compare the CV values of mean travel times for all the quarters including Q1 and excluding Q2. For instance, there are 4, 97, 15, and 68 census tract pairs which have a mean travel time of more than 240 min (4 h) during Hour 6, 7, 8, and 9, respectively (see Figure 7). As Figure 7 shows, those census tract pairs are around the central area (Boston City). Figure 8 shows the distributions of CV values of mean travel times during Hour 7 and Hour 8 for all the quarters including Q1 and excluding Q2. In Figure 8, 'A' means all the quarters including Q1, and 'B' means all the quarters excluding

Q1. 'A' has some CV values above 1, whilst the vast majority of CV values of 'B' are below 0.5. This indicates that Q1 has more outliers which are extremely high. Therefore, we average the mean travel times of Q2, Q3, and Q4 instead of those of all four quarters to represent the annual mean travel time.

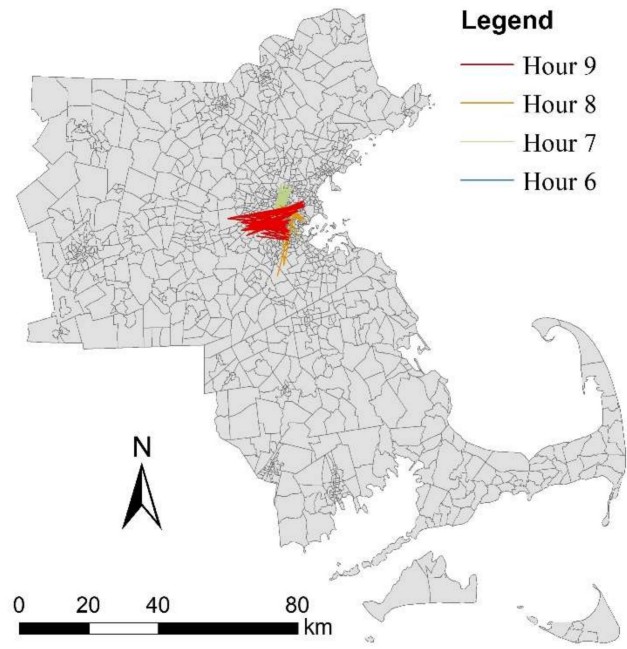

**Figure 7.** Mean census tract-to-census tract (CT-to-CT) travel times exceeding 4 h in Quarter 1.

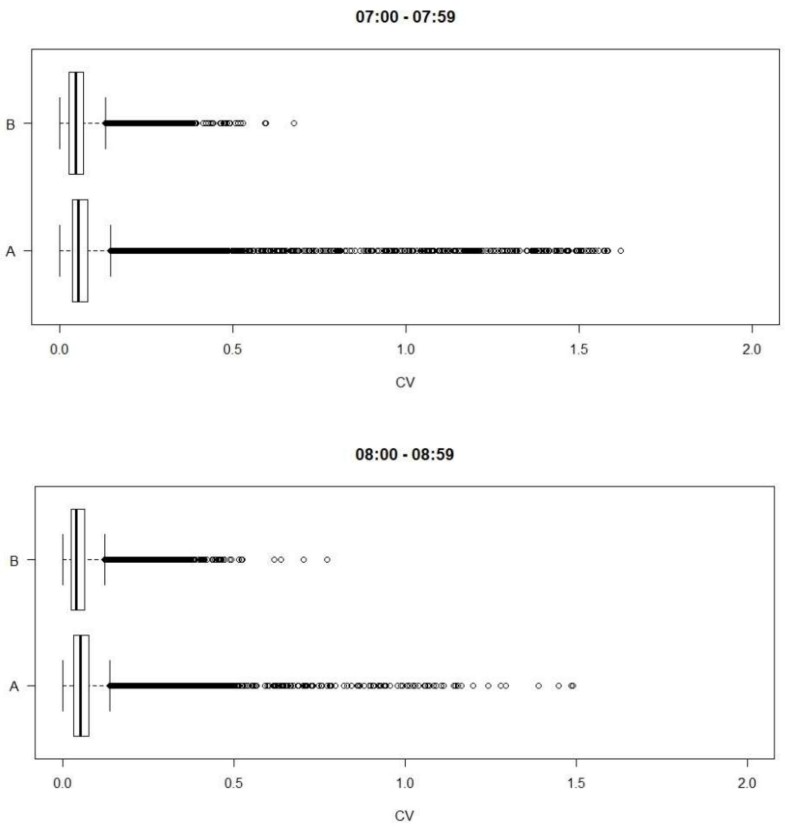

**Figure 8.** Distributions of coefficient of variation (CV) values of mean travel times during Hour 7 and Hour 8 (note: 'A' means all the quarters including Q1, and 'B' means all the quarters excluding Q1).

### 3.3. Estimation of Average Car-based Commuting Times

In this study, we used the annual mean travel times of Uber cars during morning peak time periods (i.e., Hour 7 or Hour 8) to represent average one-way car-based commuting times across Greater Boston. To calculate commuters' annual mean commuting time across Greater Boston, we first need to match O–D census tracts from the Uber Movement data and the survey data. As some pairwise O–D census tracts have neither any commuting trips nor any Uber car trips, we cannot completely match O-D census tracts from the Uber Movement data and the survey data. Table 2 shows the number of census tract pairs matched between Uber Movement and O–D flow data. Table 2 also shows the number of commuters travelling between census tract pairs matched to O–D flow data.

**Table 2.** Number of census tract pairs matched between Uber Movement and origin–destination (O–D) flow data.

|  | H7 | H8 |
|---|---|---|
| Number of census tract pairs matched to OD matrix data | 53,851 | 54,372 |
| Number of commuters travelling between census tract pairs matched to OD matrix data | 616,727 | 625,646 |

As a result, using the equations in Section 2, the averages of all car-based commuters' annual mean travel time during Hour 7 and Hour 8 are estimated to be 20.1 min and 20.5 min, respectively, in Greater Boston. More than 75% of car-based commuters are likely to have a one-way commuting time of less than 30 min. Less than 1% of car-based commuters are likely to have a one-way commuting time of more than 60 min. According to the American Community Survey (ACS), Boston City had an average commuting time of 29.3 min in 2016 [29], whilst the average commuting time by car is estimated to be around 20 min by using Uber Movement data. It seems the average commuting time by car estimated in this study (approximately 20 min) is reasonable as it should be shorter than the commuting time by all modes (29.3 min).

In this study, we need to compare the commuting times estimated from Uber Movement data and other data to somewhat validate the estimation. Specifically, we compare the commuting times by car estimated by Uber data and commuting times by all modes estimated by survey data, since commuting times by car estimated by survey data are not available (see Table 3). Table 3 also shows the commuting times estimated from the 2013–2017 American Community Survey of U.S. Census Bureau [32]. Although commuting times estimated by survey data are not available for Greater Boston, commuting times in Greater Boston should have a similar distribution to those in Boston City or Massachusetts. It could be inferred that, according to the survey data, nearly half of commuters in Greater Boston have a commuting time of less than 30 min and nearly 12% of commuters in Greater Boston have a commuting time of more than 60 min. Nevertheless, according to estimation using Uber Movement data in this study, more than 75% of commuters in Greater Boston have a commuting time of less than 30 min and less than 1% of commuters in Greater Boston have a commuting time of more than 60 min. This empirical study proves that public transport users are suffering from longer commuting times than car users. This offers new evidence on the need to enhance the public transport service in Greater Boston, since reducing inequalities in job accessibility is necessary.

Likewise, we calculate the annual mean travel times of Uber cars in 2016 across Greater Washington DC. There are 558 census tracts in Greater Washington DC. Using the equations in Section 2, the averages of all car-based commuters' annual mean travel time during Hour 7 and Hour 8 are estimated to be 21.1 min and 21.9 min, respectively, in Greater Washington DC. Table 4 shows one-way commuting time estimated across Washington DC by Uber Movement data. More than 75% of car-based commuters are likely to have a one-way commuting time of less than 30 min. Less than 1% of car-based commuters are likely to have a one-way commuting time of more than 60 min. According to the American Community Survey (ACS), the nationwide average commuting time by car is 25.4 min in 2016 [33]. The one-way

commuting time estimated in Greater Boston and Greater Washington DC are shorter than the national average level (25.4 min).

**Table 3.** Comparison of commuting time estimated by Uber data and survey data.

| | | 0–29 min | 30–59 min | 60+ min |
|---|---|---|---|---|
| Car-based commuting time estimated by Uber data | Greater Boston (07:00–07:59) | 78.77% | 21.16% | 0.07% |
| | Greater Boston (08:00–08:59) | 77.22% | 22.77% | 0.01% |
| All-mode commuting time estimated by survey data | Boston City | 47.5% | 40.7% | 11.8% |
| | Massachusetts | 54.9% | 32.8% | 12.3% |
| | United States | 62.5% | 28.6% | 8.9% |

Source: U.S. Census Bureau, 2013–2017 American Community Survey [32].

**Table 4.** One-way commuting time estimated across Washington DC by Uber Movement data.

| | | 0–29 min | 30–59 min | 60+ min |
|---|---|---|---|---|
| Car-based commuting time estimated by Uber data | Greater Washington DC (07:00–07:59) | 81.25% | 18.74% | 0.01% |
| | Greater Washington DC (08:00–08:59) | 78.4% | 21.59% | 0.01% |

*3.4. Spatial Accessibility of Jobs by Car*

Moreover, we further aggregated CT-to-CT commuting times to origin census tracts to measure spatial accessibility of jobs by car. Equation (4) was used to calculate spatial accessibility of jobs by car at the census tract level. Figure 6 shows average commuting times of car users by origin census tract. The grey areas are the census tracts without CT-to-CT commuting times. The average commuting times of car users by origin census tract can be used to represent spatial accessibility of jobs by car at the census tract level (Figure 9). In Figure 9, the census tracts with an average commuting time of more than 20 min are unlikely to cluster. This indicates that the areas suffering a lower level of spatial accessibility of jobs by car are likely to be evenly distributed across Greater Boston.

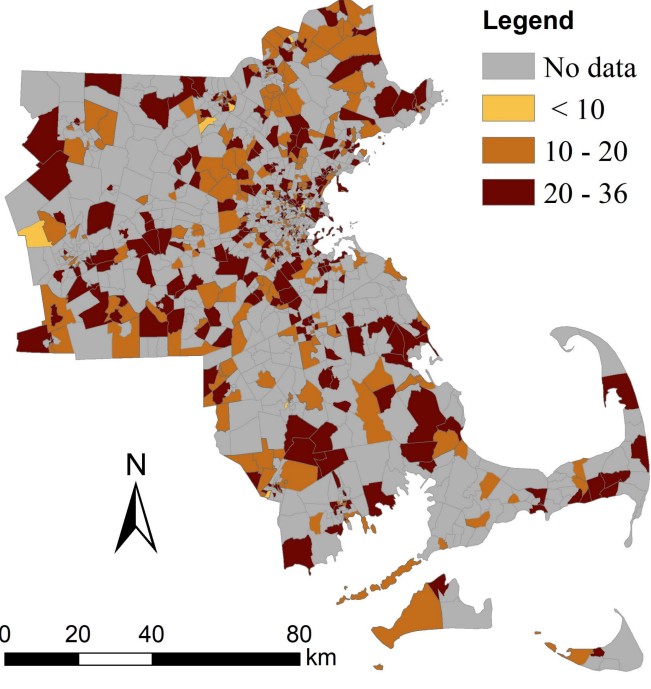

**Figure 9.** Spatial accessibility of jobs by car across Greater Boston (unit: minute).

*3.5. Discussions*

Some of the Uber trips are non-commuting trips but are not excluded by the aggregations. It is not extremely clear to what extent travel purpose would influence travel time due to the lack of empirical studies on non-commuting travel times. If the priority is given to minimizing the travel time, both commuting and non-commuting Uber trips are likely to be the fastest trips with the highest average speed. In this case, travel purpose might have a weak influence on travel time. Mean travel times of Uber cars sometimes might be not equal to real commuting times by car as real commuting times include not only in-vehicle times but time spent walking from home or office to car and parking. As the numbers of real car-based commuters between two census tracts are unknown, we use the number of all-mode commuters to represent the distribution of car-based commuters among census tract pairs based on an assumption that the proportion of car-based commuters is likely to be evenly distributed among census tract pairs. However, the real proportion of car-based commuters is unlikely to be evenly distributed. Therefore, the gap between the realistic commuter's one-way average commuting time and the estimated commuter's one-way average commuting time is worth attention and caution.

Since Uber Movement data take into account in-vehicle times but neglect the time spent walking and parking, real commuting times by car sometimes might be slightly longer than mean travel times of Uber cars. We do not use the Uber Movement data of Q1 to calculate annual travel times as it has a number of outliers. There might exist a bias when averaging travel time in three quarters to represent annual travel time. In addition, there exists a one-year gap between commuting flow data and Uber Movement data. It is unknown how the distribution of commuting flows among census tracts differed between 2015 and 2016. As Uber Movement data are only available only for three years (2016, 2017, and 2018), we cannot undertake a longitudinal study of car-based commuting times using Uber Movement data. Additionally, it seems that ride-sharing data from UberPOOL services are not included in the Uber Movement data, although the percentage of Uber rides that are on Uber Pool globally increased to 20% in 2016 [34]. As ride-hailing trips between two places are likely to be longer than car-based trips, travel times by UberPOOL services are likely to be longer than those by Uber car services, called UberX services.

In addition, as Uber Movement data exclude within-census tract travel times, we do not take into account within- census tract travel times in this study. Here, we have an issue thatnegligence of within-census tract travel times might have a substantial influence on the estimate results in this study. Based on the statistical analysis of the census tracts' areas in Greater Boston, 70% of census tracts have an area of below 9 km$^2$. If census tracts could be approximated to squares, 70% of census tracts would have a width of below 3 km. We might infer that the majority of within-census tract trips are unlikely to be taken via Uber services as their distances are likely to be below 3 km. In this case, negligence of within-census tract travel times might not have a substantial influence on the estimate results in this study.

## 4. Conclusion and Future Work

In this study, to demonstrate usefulness of Uber Movement data, we use Uber Movement data as a proxy for commuting time data by which commuters' average one-way commuting time across Greater Boston can be figured out. We propose a new approach to estimate the average car-based commuting times through combining commuting times from Uber Movement data and commuting flows from travel survey data. The empirical results further uncover that 1) commuters' average one-way commuting time is around 20 min across Greater Boston; 2) more than 75% of car-based commuters are likely to have a one-way commuting time of less than 30 min; 3) less than 1% of car-based commuters are likely to have a one-way commuting time of more than 60 min; and 4) the areas suffering a lower level of spatial accessibility of jobs by car are likely to be evenly distributed across Greater Boston.

Some limitations in the empirical study need to be noted. Firstly, this study focused on car-based commuting time rather than commuting times by all transport modes, as travel times by Uber car are close to those by private cars. Secondly, Uber offers average travel time but not the number of trips between small areas. The availability of Uber trip flow data could definitely enhance usage of Uber Movement data in transport research. Thirdly, we used Equation (3) instead of Equation (1)

to calculate average car-based commuting times by assuming that all pairwise census tracts have the same proportion of commuters who are travelling to work by car, but actually the proportion of car-based commuters is likely to change from one pair of census tracts to another. Finally, demographic characteristics of commuters are unknown, such as age, gender, race, profession, income, education and so forth. It would be more informative to incorporate demographic characteristics of commuters into this study.

In future, some further aspects should be considered in support of research enhancement. Firstly, if Uber plans to release aggregated Uber O-D matrix datasets in the near future, it is of high interest to exploit Uber O-D matrix data to complement transport research. Secondly, we might acquire demographic characteristics of Uber car riders through visiting Uber users' profiles to acquire personal information such as gender, age, profession, and so forth. Thirdly, as some cities, such as Beijing, New York City, and City of Chicago, have released conventional taxi trip datasets, it is of high interest to compare or combine conventional taxi trip data and Uber trip data in transport studies. Finally, Uber has also released average travel speed data between census tracts in a few cities and plans to extend the speed data availability to more cities worldwide. Investigating how average speed varies over space would enable a better understanding of traffic efficiency, and thus could inform urban planning and traffic monitoring.

**Author Contributions:** Conceptualization: Yeran Sun and Xuan Sun; Methodology: Yeran Sun and Xuan Sun; Software: Yeran Sun and Yinming Ren; Validation: Yeran Sun and Yinming Ren; Formal Analysis: Yeran Sun and Xuan Sun; Investigation: Yeran Sun and Xuan Sun; Resources: Yeran Sun and Yinming Ren; Data Curation: Yeran Sun and Yinming Ren; Writing-Original Draft Preparation: Yeran Sun and Xuan Sun; Writing-Review & Editing: Yeran Sun and Xuan Sun; Visualization: Yeran Sun and Yinming Ren; Supervision: Yeran Sun. All authors have read and agreed to the published version of the manuscript.

**Funding:** This work is supported by the Fundamental Research Funds for the Central Universities (Grant No. 37000-31610453), the General Program of Social Science of Tianjin (TJGL19-005), and the Fundamental Research Funds for the Central Universities (63192205).

**Acknowledgments:** It is very grateful that Uber Movement data services offer open data to researchers.

**Conflicts of Interest:** The authors declare no conflict of interest.

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
