# Peer review of "Uber Movement Data: A Proxy for Average One-way Commuting Times by Car"

_ijgi, doi:10.3390/ijgi9030184_

Round 1

Reviewer 1 Report

This research claims to design a new approach to estimate average car-based commuting times by combining commuting times from Uber Movement data and commuting flows from travel survey data.   

1.If the purpose of this study is to demonstrate the usefulness of Uber movement data in estimating the one-way car-based commuting times, I am having trouble with this specific motivation. Was this study carried out solely because of the availability of Uber movement data? 

2. The author used the annual mean travel times of Uber cars during morning peak time periods (i.e., Hour 7 or Hour 8) to represent average one-way car-based commuting times across Greater Boston. Why? I don’t agree with the ‘intuition’. 

3. I didn’t really see a novel contribution from the method proposed by author to combine travel survey data and Uber movement data. What’s added value of travel survey data? If you already have uber movement data to tell you the average car-based commuting times, I didn’t see a strong motivation for adding travel survey data, notwithstanding the fact that travel survey data also has sampling bias.  A hypothetical example, if we know that it takes a Uber car on a Weekday 06:00am 25 minutes to go from C1 to C2, why does it still matter if this Uber car is driving among a 100 cars on the road or 10000 cars on the road? Traditionally, we use the number of cars on the road to estimate the travel time between C1 and C2 as we don’t have data like Uber Movement data. 

4. If the author wants to study the uber movement data, I suggest at least the author should analyze uber movement data for multiple cities from multiple regions. Then, the author can discuss the usefulness of uber movement data in capturing human travel time.

Author Response

Reviewer 1

This research claims to design a new approach to estimate average car-based commuting times by combining commuting times from Uber Movement data and commuting flows from travel survey data.  

Response: Thank you for your constructive comments. We carefully revise our manuscript according to the comments from you and other reviewers. Moreover, as the other reviewers suggest, we add an analysis of spatial job accessibility by car. We hope you are satisfied with the revised version.

1.If the purpose of this study is to demonstrate the usefulness of Uber movement data in estimating the one-way car-based commuting times, I am having trouble with this specific motivation. Was this study carried out solely because of the availability of Uber movement data?

Response: In the revised paper, we further link the Uber movement data to spatial accessibility of job. As the other reviewers suggest, we deepen the analysis by adding an analysis of spatial job accessibility by car to link the commuting time estimates to spatial accessibility of job (see sub section 2.5 and 3.4).  

  1. The author used the annual mean travel times of Uber cars during morning peak time periods (i.e., Hour 7 or Hour 8) to represent average one-way car-based commuting times across Greater Boston. Why? I don’t agree with the ‘intuition’.

Response:

In the revised paper, we rewrite this sentence into “Uber cars’ average travel times in the morning on weekdays could be used as a proxy for average one-way (home-to-work) car-based commuting times due to the same transport mode and time periods.”

  1. I didn’t really see a novel contribution from the method proposed by author to combine travel survey data and Uber movement data. What’s added value of travel survey data? If you already have uber movement data to tell you the average car-based commuting times, I didn’t see a strong motivation for adding travel survey data, notwithstanding the fact that travel survey data also has sampling bias. A hypothetical example, if we know that it takes a Uber car on a Weekday 06:00am 25 minutes to go from C1 to C2, why does it still matter if this Uber car is driving among a 100 cars on the road or 10000 cars on the road? Traditionally, we use the number of cars on the road to estimate the travel time between C1 and C2 as we don’t have data like Uber Movement data.

Response: Incorporating travel survey data into the study is to calculate the average travel time of all commuters. A hypothetical example, if there are 10 commuters travelling from C1 to C2 with a mean time of 20 minutes, and 40 commuters from C3 to C4 with a mean time of 10 minutes, the average of travel time of the 50 commuters is (10*30 + 40*10)/(10+40) = 14 minute.

  1. If the author wants to study the uber movement data, I suggest at least the author should analyze uber movement data for multiple cities from multiple regions. Then, the author can discuss the usefulness of uber movement data in capturing human travel time.

Response:

In the revised paper, we added the analysis of another city Washington DC.

Reviewer 2 Report

This paper presents an interesting topic related to the average travel times, in different time periods, across Greater Boston. However the authors do not explore properly the spatial component of the data, fundamental for the perception of the areas of greater / lesser influx of travel. This paper need some maps illustrating the results of the temporal analysis carried out; The document also requires some investment in formatting and revising the text; Authors must avoid frequent repetition of the same objectives throughout the text; The work lacks a geographic, social and demographic contextualization of the results; The practical applicability of the results obtained is little explored by the authors.

Some issues that desserve, in my view, a better clarification in order to improve the work:

  1. The main goals of this study are well defined, however it seems to me that the authors can, with the information they have, go a little further. In addition of estimating average car-based commuting times, the classification of the study area, according to its accessibility, proposed by the described method, seems to be also very relevant for a better management of road traffic in an urban environment or even optimize the spatial distribution of the car fleet for a service like Uber. A good spatial perception of travel direction (from-to) is essencial and the inclusion of this component in this study, will allow to demonstrate, in practice, its applicability.
  2. Figure 1 does not add much information to the study, if only the limits of the census areas are represented. However, it shows some variability of the geographic extension of each census tract. Some questions arise taking this issue into account: averages of distances calculated in each census tract can be affected by their geographic extention? how?

Within the same census tract unit, point location of source or destination can be very different, especially if its geographic extension is high!

  1. I don't really see the purpose of exploring the distributions of mean CT-to-CT travel times by quarter. Can the authors explain the advantage of this analysis at this level of detail?
  2. line 190: “Hour 7 and Hour 8 are more likely to be the morning peak time periods due to higher medians of mean travel times”. In what context this conclusion is new considering the normal dynamics of a city? Isn't that an obvious conclusion?!
  3. line 196: where is described the “coefficient of variation”?
  4. line 198: “there are 4, 97, 15, and 68 census tract pairs which have a mean travel time of more than 199 minutes (4 hours) during Hour 6, 7, 8, and 9 respectively.” – Mapping this information, identifying these spatial units (census tract) it's imperative! The question, beside “how long?” is also “where?”!
  5. line 220: “the averages of car-based commuter’s annual mean travel time during Hour 7 and Hour 8 are 20.1 minutes and 20.5 minutes respectively” - What is the practical meaning of this infinite difference? does it really deserve to be mentioned? Is yes, why?!
  6. Globally the text still needs revision in english and some formatting (some example):

line 21: to estimate;

line 101:  replace "as follows." by "as follows:";

line 181 and 182: Hour 8 (07:00 – 07:59)?!;

line 245: Table 3. Comparisons;

Equation 2 requires some formatting ;

References require some formatting according to the journal graphic standards. The web links used in the bibliographic references must include the date of access;

Author Response

Reviewer 2

This paper presents an interesting topic related to the average travel times, in different time periods, across Greater Boston. However the authors do not explore properly the spatial component of the data, fundamental for the perception of the areas of greater / lesser influx of travel. This paper need some maps illustrating the results of the temporal analysis carried out; The document also requires some investment in formatting and revising the text; Authors must avoid frequent repetition of the same objectives throughout the text; The work lacks a geographic, social and demographic contextualization of the results; The practical applicability of the results obtained is little explored by the authors.

Response: Thank you for your constructive comments. We carefully revise our manuscript according to the comments from you and other reviewers. Important changes are marked in red colour to highlight. We hope you are satisfied with the revised version.

We deepen the analysis by adding an analysis of spatial job accessibility by car to link the commuting time estimates to spatial accessibility of job (see sub section 2.5 and 3.4). As you suggest, we add more maps to display the results of the temporal analysis (see Figure 5 and 6).

Some issues that deserve, in my view, a better clarification in order to improve the work:

The main goals of this study are well defined, however it seems to me that the authors can, with the information they have, go a little further. In addition of estimating average car-based commuting times, the classification of the study area, according to its accessibility, proposed by the described method, seems to be also very relevant for a better management of road traffic in an urban environment or even optimize the spatial distribution of the car fleet for a service like Uber. A good spatial perception of travel direction (from-to) is essential and the inclusion of this component in this study, will allow to demonstrate, in practice, its applicability.

Response:

In the revised paper, we deepen the analysis by adding an analysis of spatial job accessibility by car to link the commuting time estimates to spatial accessibility of job (see sub section 2.5 and 3.4).  In the added maps, we display the directions of commuting times (see Figure 5 and 6).

Figure 1 does not add much information to the study, if only the limits of the census areas are represented. However, it shows some variability of the geographic extension of each census tract. Some questions arise taking this issue into account: averages of distances calculated in each census tract can be affected by their geographic extention? how?

Within the same census tract unit, point location of source or destination can be very different, especially if its geographic extension is high!

Response:

We agree with that Figure 1 might not add much information to the study. The reason we keep this Figure is that some readers might have no idea what is the normal shape and area of a US census tract in Greater Boston region.

We totally agree that the variability of the geographic extension is likely to influence the calculation of distances between census tracts. However, we focused on travel times which are derived directly from Uber trips instead of being figured out based the distance and speed. We think variability of geographic areas is unlikely to influence travel times in this study since we don’t focus on commuting speed in this study.

We totally agree that point location of source or destination can be very different within the same census tract unit. However, as Uber Movement data exclude within- census tract travel times, we don’t take account of within- census tract travel times in this study. Here we have another issue that: if negligence of within-census tract travel times might have a substantial influence on the estimate results in this study. We add more discussions on this issue (see red words at the end of sub section 3.5. Discussions).

I don't really see the purpose of exploring the distributions of mean CT-to-CT travel times by quarter. Can the authors explain the advantage of this analysis at this level of detail?

Response: First, the data is aggregated for quarters rather than years although we really need the data aggregated for years in this study. In this case, we have to average the average travel times of the four quarters to represent the annual average travel time. Moreover, as some events such as road closures or reconstructions might influence the traffic speed in one season, we first filter out the season with less regular speed compared to other seasons before we calculate the annual average travel times based on the average travel times of the quarters.

line 190: “Hour 7 and Hour 8 are more likely to be the morning peak time periods due to higher medians of mean travel times”. In what context this conclusion is new considering the normal dynamics of a city? Isn't that an obvious conclusion?!

Response: We have this statement because one comment from another reviewer suggest us to give reasons why certain time periods are selected as morning peak in the previous round review.  

line 196: where is described the “coefficient of variation”?

Response: In the revised paper, we defined the coefficient of variation (see red words in sub section 3.2. Calculation of annual mean travel times of Uber cars).  

line 198: “there are 4, 97, 15, and 68 census tract pairs which have a mean travel time of more than 199 minutes (4 hours) during Hour 6, 7, 8, and 9 respectively.” – Mapping this information, identifying these spatial units (census tract) it's imperative! The question, beside “how long?” is also “where?”!

Response: In the revised paper, we add a map to show those spatial units (see Figure 7).

line 220: “the averages of car-based commuter’s annual mean travel time during Hour 7 and Hour 8 are 20.1 minutes and 20.5 minutes respectively” - What is the practical meaning of this infinite difference? does it really deserve to be mentioned? Is yes, why?!

Response:

In this study, average travel times of Uber cars in the morning peak time periods (Hour 7 and Hour 8) on weekdays are used as a proxy for average one-way car-based commuting times. This two figures can be used to represent approximately average one-way car-based commuting times. They show that the annual average commuting time of all car commuter is estimated to be around 20 minutes by using Uber Movement data.

Globally the text still needs revision in english and some formatting (some example):

line 21: to estimate;

line 101:  replace "as follows." by "as follows:";

line 181 and 182: Hour 8 (07:00 – 07:59)?!;

line 245: Table 3. Comparisons;

Equation 2 requires some formatting ;

References require some formatting according to the journal graphic standards. The web links used in the bibliographic references must include the date of access;

Response:

In the revised paper, we carefully proof read the paper and correct the typos as you list.  Besides, we added the date of access to the links in the references.

Reviewer 3 Report

This is a very interesting piece of work, with a novel source for data collection: Uber movement. Yet, why specifically Uber, as other current datasets, mentioned in the article, are now available e.g. Strava metro etc.? What about crowdsensing approaches?

Regarding the one-way commuting time derived from Uber data, how do we know travellers are not using the service for leisure or tourism purposes? Is this enough to conclude on the commuting patterns?

Very interesting to triangulate data sources e.g. travel survey and Uber data, to compare results.

The type of analysis presented in this paper should be provided at the start of the method section

Very nice effort in visualising Uber movement data output to contextualise the analysis, and clear presentation of the variables included in the model

References needed for ACBCT, CT-CT; avoid acronyms for non-expert reader

Not sure what’s the real value of focusing solely on commuting times and commuting flows, what about deepening the analysis on the type of travellers, sociodemographic, destination, linking it to spatial aspects? It is mentioned in conclusion/extension of the paper, but it would be worth filtering the results in a more informative way.

Author Response

Reviewer 3

This is a very interesting piece of work, with a novel source for data collection: Uber movement. Yet, why specifically Uber, as other current datasets, mentioned in the article, are now available e.g. Strava metro etc.? What about crowdsensing approaches?

Response: In the revised paper, we add more discussions on this issue (see red words in the end of 2nd paragraph of section 1 Introduction).

Regarding the one-way commuting time derived from Uber data, how do we know travellers are not using the service for leisure or tourism purposes? Is this enough to conclude on the commuting patterns?

Response:

To make trips aggregated in the Uber movement data be representative of commuting trips, we selected the data aggregated from Uber trips in weekday morning to avoid the majority of non-commuting trips. Although there might be still some non-commuting trips in weekday morning, the travel time between same origin and destination places might not differ much from commuting trips and non-commuting trips since the shortest or fastest routes are always the optimal routes selected by both commuters and non-commuters. Compared to travel mode and origin-destination, trip purpose is likely to have a relatively weak influence on travel times. Since we can theoretically assume that Uber car driving speed is close to private car driving speed between the same origin and destination places during the same time periods, we attempt to use Uber Movement data as proxy for one-way commuting times by car (see the red words in the end of section 1).

In the revised paper, we add more discussions on this issue (see the red words in sub section 3.5. Discussions).

Very interesting to triangulate data sources e.g. travel survey and Uber data, to compare results.

Response: Many thanks.

The type of analysis presented in this paper should be provided at the start of the method section

Response: We present the analysis in general at the start of the method section (see the 1st paragraph of section 2: Materials and Methods). Besides, we think it is better to introduce the data before the estimate methods as the estimate methods are selected according to the characteristics of Uber data and travel survey data.

Very nice effort in visualising Uber movement data output to contextualise the analysis, and clear presentation of the variables included in the model

Response: Many thanks.

References needed for ACBCT, CT-CT; avoid acronyms for non-expert reader

Response: In the revised paper, we added references to CT-CT and deleted ACBCT (see sub section 2.1.3. Travel survey data: commuting flow data).

Not sure what’s the real value of focusing solely on commuting times and commuting flows, what about deepening the analysis on the type of travellers, sociodemographic, destination, linking it to spatial aspects? It is mentioned in conclusion/extension of the paper, but it would be worth filtering the results in a more informative way.

Response: In the revised paper, we deepen the analysis by adding an analysis of spatial job accessibility by car to link the commuting time estimates to spatial accessibility of job (see sub section 2.5 and 3.4).  

Round 2

Reviewer 1 Report

My overall opinion for this manuscript is not enough scientific contribution. Please consider redesigning this study. 

Author Response

Thank you very much for your comments. We appreciate your advices and suggestions. In the latest version, we redesign the study and add the spatial accessibility analysis part. We hope you are satisfied with the paper.

Reviewer 2 Report

I would like to thank the authors for having accepted my suggestions in presenting a geographic context of the results and a direct application of this study. The analysis and discussion of the results is adequate to the changes made. The text is better written.

I think that the changes made gave to the study greater quality; The 1st version was too focused in the temporal component, being translated only by box-plots, and one good picture is worth a thousand words.

minor changes:
1. Line 206: Hour 8 (8:00 am - 8:59 am)
2. a review should be made of the text font used in the titles of the chapters and sub-chapters;

Congratulations to the authors for the improvement effort.

Author Response

Thank you very much for your instructive comments. We appreciate your advices and suggestions which guide us to enhance the paper. We made the further corrections according to your minor change comments. 

This manuscript is a resubmission of an earlier submission. The following is a list of the peer review reports and author responses from that submission.

Round 1

Reviewer 1 Report

This research tries to demonstrate a new approach to accurately estimate car-based average commuting times, and how average car-based commuting time changes across Greater Boston.  

The concept of “average travel time of a geographic region R” is ill-defined. It’s very closely related with the spatial scale of the geographic region, the public transit infrastructure of the region and possibly the weather conditions. I fail to see the true value of studying the average travel time of a single region without providing any verification.  What’s the uncertainty of estimation result? How can we verify that the estimation is actually realistic? Moreover, the seasonal variation presented in the article lacks any explanation. What are the possible reasons for this seasonal variation?  The method proposed by author to combine travel survey data and Uber movement data lacks scientific reasoning. What’s added value of travel survey data? If you already have uber movement data to tell you the average car-based commuting times, I didn’t see a strong motivation for adding travel survey data, notwithstanding the fact that travel survey data also has sampling bias.  A hypothetical example, if we know that it takes a Uber car on a Weekday 06:00am 25 minutes to go from C1 to C2, why does it still matter if this Uber car is driving among a 100 cars on the road or 10000 cars on the road? Traditionally, we use the number of cars on the road to estimate the travel time between C1 and C2 as we don’t have data like Uber Movement data. Now we have data like Uber movement data, then it feels like circular reasoning to add travel survey data.  Why would you assume the uber average travel time between 6:00am ~6:59am is the closest to reality?

Reviewer 2 Report

In this paper, the authors link a Uber dataset from Boston to mobility survey data, in order to calculate mean commuting times and investigate seasonal and spatial variations.

In total, the paper in its current form is not suitable for publication in IJGIS. I would recommend rewriting the manuscript and addressing the following issues:

The manuscript requires thorough proofreading (e.g., the sentence in L21 ff. contains several mistakes: “In this study, we aim to demonstrate a new approach to accurate estimates of car-based average commuting times …”).

Referencing a bunch of papers, such as in L41 [1-8] or L53 [15-22], is not acceptable. To which aspect in which paper do you refer to? You cannot expect the reader to work through 9 very different papers, each dealing with specific problems, and search for the link to your statement. I am left with the impression of name/paper dropping.

In the introduction, the huge body of literature on floating car data is missing. You should definitely have a look at it.

The central argument of this paper is introduced in L74. However, any argument for using Uber data as proxy for commuting times, except for intuition, is missing.

In L84 ff., you argue that flows can be derived from survey data, while the Uber data set facilitates the extraction of travel times. I do not agree, but would argue that the number of trips and the trip purpose is typically derived from mobility surveys and flows (spatial and temporal distribution of trips) are extracted from movement datasets (trajectories or aggregates).

The introduction does not properly reflect the problem at a conceptual level. The underlying question is how very different data sources can be linked to each other based on a common geographical reference.

Based on Figure 1, I guess that Uber data were downloaded for 2016. The travel survey data are from 2015, according to section 2.1.2. Any discussion of potential implications of this time lag are missing.

I read the sentence in L134 ff. multiple times, but was unable to understand the boxplot in Figure 2. Please elaborate on what the statistics and the chart tell us.

The average commuting times for the different regions requires additional explanation. Do the number of commutes refer to sources, destinations or any transit?

In L218 ff., you state that the analysis of the commuting trips is limited to one hour in the early morning. Do you have any indication that this is the morning peak hour in the Boston region? If yes, provide the reference.

Figure 4: please improve the design of this map. The super large compass rose has no purpose at all. Instead, use an adequate base map for improving orientation. The same holds true for the rest of the maps.

The segmentation approach of the Boston area leads to disconnected districts (section 3.2). Please add an explanation for this pattern and discuss the implications for the subsequent analyses.

Section 3.4: the variation of commuting times across seasons and regions need to be investigated statistically, in order to test for significance of change.

A thorough discussion of this research is missing. Thus, it remains unclear to which degree the research questions could be answered and how this research contributes to the existing body of literature.

Information in L315 ff. is missing.

Reviewer 3 Report

Manuscript Number: ijgi-649189

Title: Combining Uber movement data and travel survey data to estimate average car-based commuting times: a case study in Greater Boston

Review: This article uses Uber datasets containing information about average travel times and travel survey data in order to estimate zonal variations of commuting times by car in Boston (USA). The results show that specific areas present higher average commuting times and relatively high seasonal differences.

Overall view: the research presents two main contributions: the use of new data from the Uber movement dataset to estimate commuting trip times and the zonal division of the study area applying a community detection algorithm. However, the analysis of the results is relatively poor and only the differences in the average commuting times among zones and among quarters are highlighted. These differences are not explained in the text, and the conclusions focus too much in the possibilities of this new data source for future research and not in the contribution of the present paper. Therefore, in my opinion the lack of contributions in the paper justifies the rejection for publication in IJGI.

Major comments:

Introduction. The authors should discuss in more detail other contributions related to the calculation of mobility and accessibility indicators using new data sources. For example, other researchers have used TomTom data to estimate dynamic accessibility indicators to different opportunities. However, the authors seem to ignore these contributions focusing only on the data available from companies such as Strava or Uber.

Table 3. The communities C1-C3 are very small and have a low or null number of within commuters. The authors should clarify this point and check if the division in communities proposed is correct.

Section 3.3 and 3.4. There is not a clear reason why commuting trip time by car is higher in community C5. Are the transport infrastructures or the congestion higher in this area of Boston? In addition, the discussion section does not contrast the results obtained with those from other authors.

Minor comments:

Pg 7, line 221. The meaning of the acronym “CTs” should be included in the text.

Pg 8, line 246. The format of this paragraph should be corrected.

References. All the references are numbered two times.

Sections “Authors contributions”, “Funding”, “Acknowledgments” and “Conflicts of Interest” do not contain the requested information.

Reference 23 should be completed, for example including a link to the description of this new source of data.

Reviewer 4 Report

This paper aims to investigate seasonal and regional variations of average commuting time by a car while utilizing both Uber data and conventional travel data. So, the authors attempt to offer a more accurate approach to estimating travel time and transport accessibility. Using Uber and transport data from great Boston, they emphasize the importance of 1) seasonal trend in average car travel-time and 2) the variety of travel-time in different geographical spaces.

After reading it, I do remain somewhat puzzled about what the contribution of this paper to the scientific literature is. Still, several things should be improved in this paper.

The most important issues with the current version of the paper are as follows:

My most concern is related to data in terms of both Uber movement data and Travel survey data. The authors have used data that related to the average travel time of Uber riders between origins and destinations in census tract during a one-hour period on weekdays between 6:00 am 6:59 am in Greater Boston. they found that the range of travel time is likely to be 10 to 22 minutes. On the other hand, the O-D commuting flows data were downloaded for the commuting trips for the workers in Greater Boston. How they can only rely on travel data in only one-hour weekdays from 6 am to 7 am. This period time cannot be identified as the morning peak hour since most of the trip destinations such as offices or schools start to work from 8 to 9 am. So, the first issue in travel time is related to the time of the day. This period of time seems to have the least Uber trips as well. I suggest that authors consider the travel time at different peak hours during weekdays and weekends. Secondly, it seems that the Uber data is indicated the travel times between ODs for all types of trips, while the travel survey data offer the OD commuting flows. So, travel time from these two sources is not comparable in terms of trip purposes and destinations. Moreover, it seems that the Uber movement data is only extracted from Uber trips. What about Uber pool trips. Considering ridesharing uber pool trips may result in larger average travel times among ride-hailing trips. The authors have estimated the average car-based commuting distance from a formula with the uncited source. How did they find that this formula is the best approach for estimating average trip distance for both car-based commuting trips and User-based non-commuting trips? The literature must be updated in terms of the evaluation of travel time by ride-hailing services and cars. The research methods used in the paper should be clearly described, including research design, data sources, and analysis process and results in presentation. The figures do not have an accurate legend based on different regions. The conclusion must contain the contribution of the study in enhancing the travel accessibility since the authors have mentioned in the introduction of the paper that they contribute to enhancing travel accessibility through investigating the differences between travel time.